# Federated Cross Learning for Medical Image Segmentation

**Xuanang Xu**[1]                                                                    XUX12@RPI.EDU
[1] *Department of Biomedical Engineering and Center for Biotechnology and Interdisciplinary Studies, Rensselaer Polytechnic Institute, 110 8th St, Troy, NY 12180, USA*

**Hannah H. Deng**[2]                                                  HDENG@HOUSTONMETHODIST.ORG
[2] *Department of Oral and Maxillofacial Surgery, Houston Methodist Research Institute, 6560 Fannin St, Houston, TX 77030, USA*

**Tianyi Chen**[3]                                                              CHENT18@RPI.EDU
[3] *Department of Electrical, Computer, and Systems Engineering, Rensselaer Polytechnic Institute, 110 8th St, Troy, NY 12180, USA*

**Tianshu Kuang**[2]                                                 TKUANG@HOUSTONMETHODIST.ORG
**Joshua C. Barber**[2]                                             JCBARBER@HOUSTONMETHODIST.ORG
**Daeseung Kim**[2]                                                    DKIM@HOUSTONMETHODIST.ORG
**Jaime Gateno**[2,4]                                                JGATENO@HOUSTONMETHODIST.ORG
[4] *Department of Surgery (Oral and Maxillofacial Surgery), Weill Medical College, Cornell University, 407 E 61st St, New York, NY 10065, USA*

**James J. Xia**[2,4]                                                  JXIA@HOUSTONMETHODIST.ORG
**Pingkun Yan**[1]                                                               YANP2@RPI.EDU

**Editors:** Accepted for publication at MIDL 2023

## Abstract

Federated learning (FL) can collaboratively train deep learning models using isolated patient data owned by different hospitals for various clinical applications, including medical image segmentation. However, a major problem of FL is its performance degradation when dealing with data that are not independently and identically distributed (non-iid), which is often the case in medical images. In this paper, we first conduct a theoretical analysis on the FL algorithm to reveal the problem of model aggregation during training on non-iid data. With the insights gained through the analysis, we propose a simple yet effective method, federated cross learning (FedCross), to tackle this challenging problem. Unlike the conventional FL methods that combine multiple individually trained local models on a server node, our FedCross sequentially trains the global model across different clients in a round-robin manner, and thus the entire training procedure does not involve any model aggregation steps. To further improve its performance to be comparable with the centralized learning method, we combine the FedCross with an ensemble learning mechanism to compose a federated cross ensemble learning (FedCrossEns) method. Finally, we conduct extensive experiments using a set of public datasets. The experimental results show that the proposed FedCross training strategy outperforms the mainstream FL methods on non-iid data. In addition to improving the segmentation performance, our FedCrossEns can further provide a quantitative estimation of the model uncertainty, demonstrating the effectiveness and clinical significance of our designs. Source code is publicly available at https://github.com/DIAL-RPI/FedCross.

**Keywords:** Federated learning, non-iid data, medical image segmentation, ensemble mechanism.

## 1. Introduction

Federated learning (FL) (McMahan et al., 2017) is an emerging decentralized learning paradigm allowing multiple data owners to collaboratively train deep learning (DL) models without sharing the raw data. Prior works (Li et al., 2019; Liu et al., 2021; Roth et al., 2021; Sarma et al., 2021; Xia et al., 2021; Yang et al., 2021; Feng et al., 2022; Xu and Yan, 2022) have demonstrated the feasibility of FL, specifically the federated averaging (FedAvg) algorithm (McMahan et al., 2017), in medical image segmentation. However, a major concern of FL is that its performance degrades when handling the data that are not independently and identically distributed (non-iid). Due to the non-convex nature of the training objective of deep neural networks, averaging the locally trained models may lead to sub-optimal solutions in the parameter space and thus cause performance degradation (McMahan et al., 2017). This phenomenon is especially prominent when the local datasets exhibit significant distribution variation from each other, which is often true for medical imaging data acquired by different clinical sites. Recent attempts tackled the non-iid problem using more advanced model optimization or aggregation strategies. Li et al. (Li et al., 2020) derived a FedProx algorithm from the FedAvg by adding extra constraints on the local optimization objective, which regularizes the local model updates in small magnitude and thus mitigates the gradient disparities caused by the non-iid data from different clients. Li et al. (Li et al., 2021) proposed a FedBN method to handle the FL non-iid problem in feature space. They updated the batch normalization (Ioffe and Szegedy, 2015) layers of all the client models locally without global aggregation. Their objective is to align the features extracted from the non-iid data in a shared embedding space. Recently, Roth et al. (Roth et al., 2021) leveraged the neural architecture search technique to build and optimize a super network, which can be seen as an ensemble of a set of sub-networks individually adapted to different client datasets. However, in the above works, the operation of averaging parameters is persistently involved, which may be subject to the same risk of performance degradation on non-iid data as the FedAvg.

In this study, we propose a novel federated cross learning (FedCross) strategy to tackle the FL non-iid problem from a new perspective, based on the insights gained through our theoretical analysis presented in Sec. 2.2. Instead of simultaneously training multiple models on different clients, our FedCross sequentially trains the model across different clients in a round-robin manner. In addition, in order to reach a comparable segmentation accuracy and also a quantitative estimation of the model uncertainty that can only be reached using centralized learning methods, we further develop a federated cross ensemble learning (FedCrossEns) method by concurrently training and deploying multiple models using our FedCross strategy. Finally, the performance of both FedCross and FedCrossEns methods was comprehensively evaluated using public datasets.

The main contributions of our work are three-fold: 1) We conduct a theoretical analysis on the FedAvg algorithm to reveal the cause of the non-iid problem in FL. 2) Based on the insights gained through the analysis, we propose a simple and yet effective training strategy, FedCross, to tackle the challenging non-iid problem in FL for medical image segmentation. Benefitting from the aggregation-free design, our FedCross can provide better performance on the non-iid data than other FL methods, especially when the local training goes through more epochs. 3) The performance of FedCross is further improved by combining it with

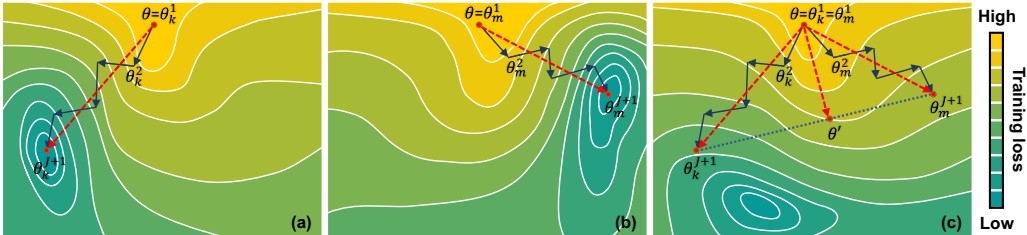

Figure 1: Model aggregation results in sub-optimal performance on non-iid data. (a) and (b) show that the locally trained models $\theta_k^{J+1}$ and $\theta_m^{J+1}$ each individually achieve their minimal in the loss landscape of the client dataset $D_k$ and $D_m$, respectively. (c) indicates that the aggregated FL model $\theta'$ is located at a non-minimal position in the global loss landscape of $D_k \cup D_m$.

ensemble mechanism, coined as FedCrossEns. FedCrossEns can provide not only a comparable segmentation accuracy in comparison to the centralized learning method, but also a quantitative estimation of the model uncertainty.

## 2. Method

In this section, we first introduce the preliminary about FL and the benchmarking FedAvg algorithm (Sec. 2.1). We then give a theoretical analysis of FL on the non-iid problem (Sec. 2.2). Motivated by our observation and analysis, we propose a new FedCross training strategy (Sec. 2.3) and combine it with the ensemble mechanism to achieve the final presented FedCrossEns method for medical image segmentation (Sec. 2.4).

### 2.1. Preliminary definition

We consider a federation consisting of a server node and $K$ client nodes. The server takes the responsibility for coordinating the communication and computation distributed in different clients, while the clients work on model training using their local data and devices. Each client $C_k$ represents an individual hospital possessing a local dataset $D_k$ that cannot be shared with other clients and the server. The goal of FL is to utilize these isolated datasets $D_k$ to collaboratively train a global model $\theta$ (*e.g.*, a segmentation network), which can provide higher accuracy than the model locally trained using a single client dataset. The conventional FedAvg algorithm iteratively achieves this goal by repeating multiple rounds of communications between the server and clients. During each communication round, the client $C_k$ individually trains a copy of the global model $\theta$ using the local dataset $D_k$ and then sends back the locally trained model $\theta_k$ to the server. The server node aggregates all client models into a new global model $\theta'$ through a parameter-wise weighted averaging $\theta' = \sum_k^K w_k \theta_k$, where the weight of client $C_k$ is determined by the size of the local dataset $\|D_k\|$ w.r.t. the whole data size $w_k = \|D_k\| / \sum_i^K \|D_i\|$.

## 2.2. Theoretical challenge of the non-iid data in FL

Let $l(\theta, D)$ be a loss function that quantifies the difference between the model output and the targets in the dataset $D$. Then a general formulation of the FL problem is to find the model parameter $\theta$ minimizing the global objective function: $\min_\theta L(\theta) = \sum_{k=1}^{K} w_k l(\theta, D_k)$. Denote the model parameter of client $C_k$ at the $j$-th local iteration as $\theta_k^j$, then the local updates of client $C_k$ can be written as

$$\theta_k^{j+1} = \theta_k^j - \alpha_j \nabla l(\theta_k^j, D_k), \text{ for } j = 1, 2, ..., J \quad \text{with} \quad \theta_k^1 = \theta \tag{1}$$

where $\alpha_j$ denotes the $j$-th step size without loss of generality, and $\theta$ is the initial global model parameter broadcast to each client at the beginning of this communication round. After all clients have finished $J$ steps of local updates, the server then obtains a new global parameter $\theta'$ via aggregating the local models $\theta_k^{J+1}$ as follows

$$\theta' = \sum_{k=1}^{K} w_k \theta_k^{J+1}. \tag{2}$$

Plugging the local update (Eq. 1) into the above equation and unraveling for $j$=$J$, $J$-1, ..., 1 will yield

$$\theta' = \theta - \sum_{k=1}^{K} w_k \alpha_1 \nabla l(\theta, D_k) - \sum_{k=1}^{K} w_k \sum_{j=2}^{J} \alpha_j \nabla l(\theta_k^j, D_k). \tag{3}$$

Observing that in the above equation, the second term on the right side is essentially $\nabla L(\theta)$ — a descent direction of the global objective function $L(\theta)$, while the third term may not be a descent direction due to the discrepancy among $\theta_k^j$ across $C_k$. As illustrated in Fig. 1, two local models each independently achieve their minimum in the loss landscape of the corresponding client dataset, but their aggregated model may fall in a non-minimal position in the global loss landscape. When the client datasets are non-iid, the discrepancy can grow arbitrarily as step $J$ increases, and thus the quality of the updated model $\theta'$ is difficult to evaluate without additional assumptions. Some variants of the FedAvg algorithm tried to tackle this non-iid problem by mitigating the discrepancy among $\theta_k^j$ across $C_k$ (such as FedProx (Li et al., 2020) and FedBN (Li et al., 2021)). However, since the model aggregation step (Eq. 2) executes during training, there is always a risk of getting low-quality $\theta'$, which may finally lead to the performance degradation on global data. When the local training goes through for a large number of steps $J$, this phenomenon is particularly evident, which can be seen from our experimental results in Sec. 3.3. Inspired by the above observation, we explore to tackle the FL non-iid problem by evading the model aggregation step during training, which results in the proposed FedCross training strategy as follows.

## 2.3. Federated cross learning (FedCross)

Unlike the conventional FedAvg (Fig. 2a) that simultaneously trains the model on multiple clients, our proposed FedCross (Fig. 2b) trains the global model solely on one client during each communication round. Specifically, in the $t$-th communication round, the server randomly selects a client $C_k^t$ as the active node for this round of training and then sends the global model $\theta^t$ to that node, which locally adapts the global model to the local dataset

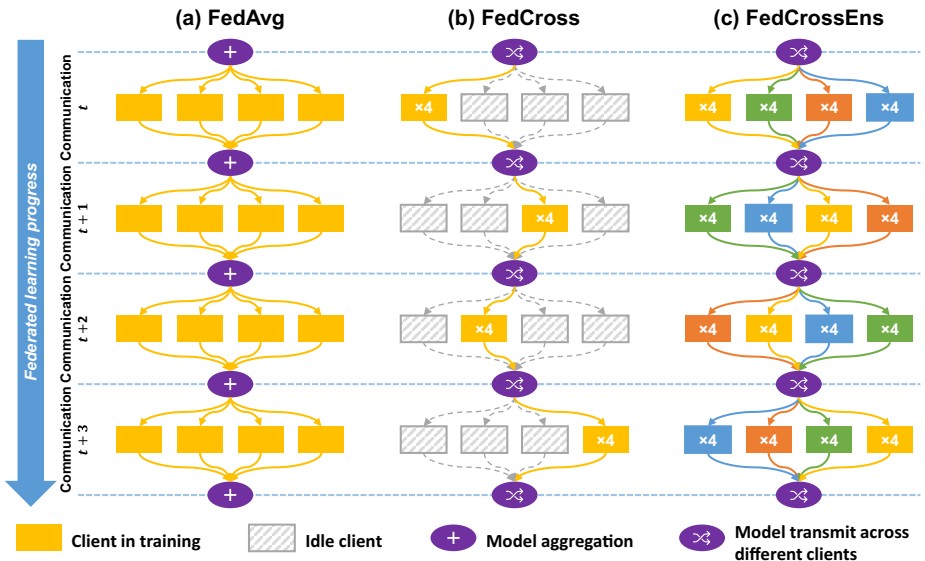

Figure 2: Training schemes of (a) FedAvg, (b) FedCross, and (c) FedCrossEns.

$D_k$. To ensure the model get equivalent training iterations on each client dataset as that in the original FedAvg algorithm, the local training epoch number is increased by $K$ times (such as the "×4" labeled in Fig. 2b). Once the training is completed, the client $C_k^t$ sends the trained model with parameter $\theta^{t+1}$ back to the server. The server then forwards it to another randomly picked client $C_k^{t+1}$ for the next round of training. By repeating the above procedure, the global model can be trained over all the client datasets without any model aggregation. From the viewpoint of a client, there is no extra communicational overhead in our FedCross when compared with the conventional FedAvg algorithm. Notably, the Fed-Cross training strategy can be seen as a special case of centralized learning, during which the training samples are grouped by different clients and fed to the model.

## 2.4. Federated cross ensemble learning (FedCrossEns)

To further boost the accuracy of our FedCross to a comparable level of the centralized learning models, we combine it with an ensemble mechanism to compose the FedCrossEns method shown in Fig. 2c. In this method, multiple models are simultaneously trained using our FedCross strategy and deployed in parallel to make ensemble predictions. Specifically, at the beginning of training, $K$ local models are initialized. Each of them is individually trained through a random route (indicated by different colors in Fig. 2c) across the clients. At the end of the training, we can get $K$ different models for the same task. We then employ an ensemble mechanism to average the predictions of the $K$ models to get a final prediction, which is expected to be more accurate than that produced by an individual model (Sagi and Rokach, 2018). In addition, we calculate the pixel-wise standard deviation of the predicted segmentation masks from these sibling models to estimate an uncertainty map, which is of great interest of clinical applications (Wang et al., 2019; Balagopal et al., 2021; Xu et al., 2022).

## 3. Experiments

### 3.1. Dataset and implementation details

The effectiveness of our method for segmentation was evaluated using four public magnetic resonance imaging (MRI) datasets for whole prostate gland segmentation, including: MSD (Antonelli et al., 2021), NCI-ISBI (Nicholas Bloch), PROMISE12 (Litjens et al., 2014), and PROSTATEx (Armato et al., 2018). The four datasets were originally collected by different clinics using different MRI scanners. Thus, they are treated as four client nodes in one federation to mimic a typical non-iid data scenario. Each of the four datasets was split into training (60%), validation (10%), and testing (30%) sets. Please refer to Table 2 and 3 in Appendix for more details regarding the data. We employed 3D U-Net (Ronneberger et al., 2015) for segmentation, which was trained using an SGD optimizer for a total of 400 epochs. For the FL methods, the 400 epochs were evenly divided into $T=\{400, 200, 100, 50, 25\}$ rounds of communications; each round contained $E=\{1, 2, 4, 8, 16\}$ local training epochs, respectively. The learning rate was initialized as 0.01 and decayed throughout the training following the poly learning rate policy (Isensee et al., 2021). The total training loss is the sum of the cross-entropy loss and Dice loss (Milletari et al., 2016) between the predicted and ground-truth segmentations. All the images were resampled to a uniform spatial resolution of $0.5\times0.5\times1.0mm^3$ and the pixel intensities were rescaled by z-score normalization. As a data augmentation strategy, we randomly cropped patches in the size of $160\times160\times32$ from each MRI volume to train the network. The training batch size was set to 16. Source code is publicly available at https://github.com/DIAL-RPI/FedCross.

### 3.2. Experimental design

In this section, we successively performed two experiments to evaluate the performance of our method and justify the effectiveness of our design, respectively. The first experiment was to evaluate the accuracy of our method by comparing it to three benchmarking training strategies, including: 1) Localized training using a single client training set to train each local client model; 2) Centralized training using a combination of all the training sets to train one global model; and 3) Federated training using all client training sets without data sharing to collaboratively train one global model. In federated training, we further compared our method with the other three representative FL methods, including FedAvg (McMahan et al., 2017), FedProx (Li et al., 2020), and FedBN (Li et al., 2021). To ensure the competing FL methods achieving their best performance, the local training epoch number $E$ was set to 1. The trained models were evaluated individually on four client testing sets using Dice similarity coefficient (DSC). Finally, the global performance was evaluated by averaging the mean DSCs and average symmetric surface distances (ASDs) across the four clients. Instead of performing an overall averaging, this strategy was to avoid the bias from unbalanced sample sizes among the clients, ensuring the global DSC and ASD calculated from each client with different sample size play an equal role. Finally, paired $t$-tests were performed for evaluating the statistical significance between FedCrossEns and other methods.

The second experiment was to evaluate the impact of local training epoch number $E$. As mentioned in Sec. 2.2, there is a risk of getting low-quality models via model aggregation in the FedAvg algorithm and its variants. Our analysis shows that this risk can be amplified

Table 1: Prostate MRI segmentation results achieved with a 3D U-Net trained using different learning strategies. Asterisk "*" indicates a result with statistical significance compared with the bottom row ($p<0.05$).

| Datasets | MSD | NCI-ISBI | PROMISE12 | PROSTATEx | Global | |
|---|---|---|---|---|---|---|
| | DSC [Mean$_{(SD)}$%] | | | | DSC [%] | ASD [mm] |
| Localized | $83.97_{(11.92)}$ | $84.04^*_{(4.87)}$ | $81.64_{(14.05)}$ | $\mathbf{90.67}_{(2.79)}$ | 85.08 | 2.33 |
| Centralized | $90.68_{(2.40)}$ | $\mathbf{87.19}_{(4.68)}$ | $86.15_{(5.56)}$ | $90.44_{(2.74)}$ | 88.61 | 1.43 |
| FedAvg | $89.96^*_{(2.85)}$ | $84.93^*_{(7.59)}$ | $81.64^*_{(9.94)}$ | $90.34^*_{(2.96)}$ | 86.72 | 1.57 |
| FedProx | $90.16_{(2.40)}$ | $86.27_{(5.02)}$ | $83.53^*_{(8.48)}$ | $90.59_{(2.84)}$ | 87.64 | 1.54 |
| FedBN | $90.06_{(2.99)}$ | $86.06_{(6.33)}$ | $83.08^*_{(7.80)}$ | $90.38_{(2.96)}$ | 87.40 | 1.58 |
| FedCross | $90.31_{(2.36)}$ | $85.96_{(6.87)}$ | $85.09_{(5.68)}$ | $90.29^*_{(2.85)}$ | 87.91 | 1.57 |
| FedCrossEns | $\mathbf{90.77}_{(2.47)}$ | $86.78_{(6.60)}$ | $\mathbf{86.72}_{(5.54)}$ | $90.66_{(2.80)}$ | $\mathbf{88.73}$ | $\mathbf{1.22}$ |

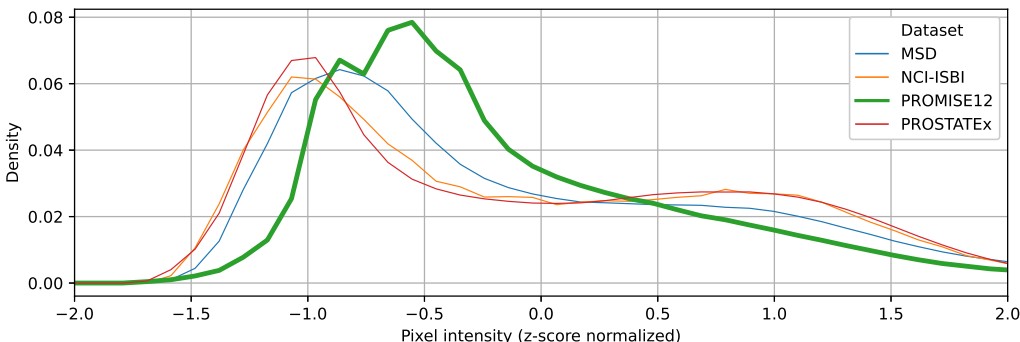

Figure 3: Histogram of the four prostate MRI segmentation datasets used in this study. PROMISE12 dataset exhibits a significant distribution shift from the other three datasets.

when the local training step number $J$ increases in Eq. 3. In this experiment, we used the local training epoch number $E$ as a surrogate of $J$, and successively evaluated the performance of different FL methods with $E=\{1, 2, 4, 8, 16\}$. For a fair comparison, we used the FedCross without ensemble mechanism.

### 3.3. Experimental results and discussion

The results of the first experiment are summarized in Table 1. Overall, our FedCrossEns achieved the highest accuracy. In addition, our FedCross also outperformed other FL methods and the localized training method even without the booster of ensemble mechanism, demonstrating the effectiveness of our FedCross strategy in tackling the non-iid data problem in FL. This was especially true on the PROMISE12 dataset whose distribution was significantly different from the other three. (Fig. 3 shows an evidence of the significantly skewed distribution.) Moreover, the ensemble mechanism helped bridge the gap between

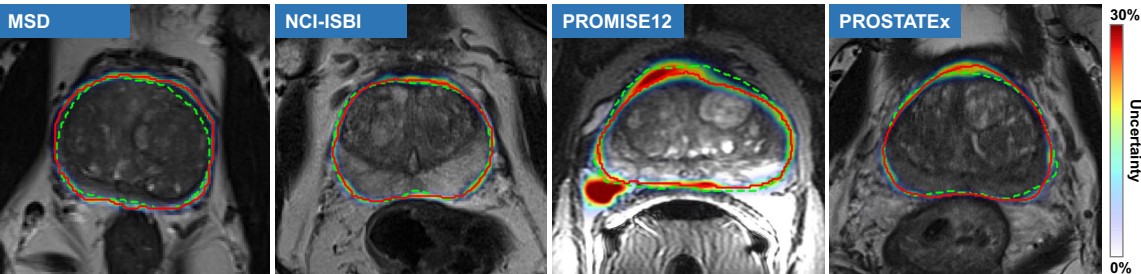

Figure 4: Ground-truth (dashed green lines) and predicted (solid red lines) segmentation superimposed with the estimated uncertainty maps.

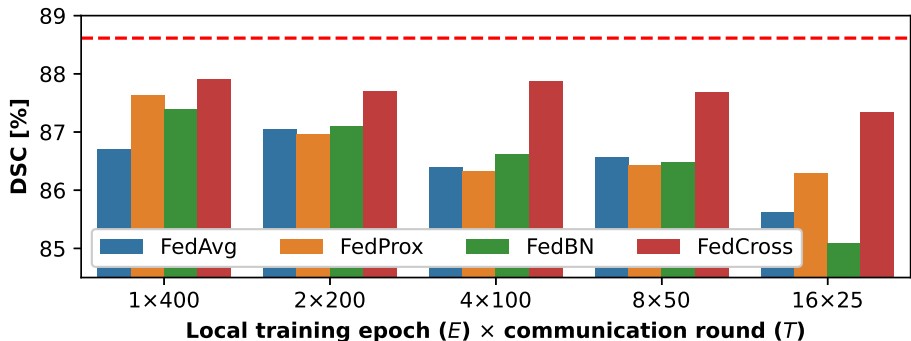

Figure 5: Segmentation accuracy of four FL methods trained with different numbers of local epochs. Red dashed line indicates the performance of centralized learning model.

the centralized learning method and our FedCross. In addition to the accuracy improvement, our FedCrossEns also estimated a pixel-wise uncertainty map of the generated contour (Fig. 4), which could in turn provide an essential guidance for the physicians in the manual modification or approval procedures. Finally, comparing to the localized learning method, all FL methods showed a significant improvement on the first three datasets with relatively small sample size (19, 48, and 30 v.s. 122 subjects in the fourth dataset), indicating its effectiveness of utilizing the knowledge contained in isolated data to collaboratively train a powerful DL model.

The results of the second experiment are shown in Fig. 5. The FL methods involving the model aggregation step (*i.e.*, FedAvg, FedProx, and FedBN) generally suffered a more significant performance degradation than our FedCross when the local training epoch number $E$ increased from 1 to 16. This result confirmed our theoretical analysis that the errors caused by the parameter-wise averaging (Eq. 2) may amplify when the local training goes through more steps. In contrast, because our FedCross strategy does not involve any model aggregation during training, the increase of local training epoch number showed a less impact on our FedCross model. Since a larger number of local training epochs indicates fewer communication rounds between the server and clients, our method may improve the efficiency and stability of an FL system.

## 4. Conclusion

This study proposes a novel FedCross training strategy to address the challenging problem of non-iid data in FL-based medical image segmentation. Unlike conventional FL methods, such as FedAvg and its variants, FedCross is designed to be aggregation-free, which can result in a better and more stable performance, especially when local training runs for a large number of epochs. Building upon the success of FedCross, our FedCrossEns method further enhances the accuracy of the FL models, which can be comparable to those of centralized learning models. Furthermore, the proposed FedCrossEns can estimate uncertainty maps for the segmentation results, as shown in Fig. 4, which provide an essential guidance for the physicians to modify or approve the predicted segmentation. That substantially increases the clinical applicability of our method.

Catastrophic forgetting and lack of parallelizability (Li and Hoiem, 2017; Sheller et al., 2020; Remedios et al., 2020) could be potential limitations of the proposed method. Since our method conducted the model training in a round-robin manner across different clients, there is a risk that the trained model may not maintain consistent performance on the client dataset that they have not experienced for a long time. Fortunately, in medical imaging-related applications, the number of clients in a federation is generally small, which may limit the impact of this catastrophic forgetting issue.

## Acknowledgments

This work was supported in part by the National Science Foundation (NSF) under CA-REER award OAC 2046708 and the National Institutes of Health (NIH) under awards R01DE022676 and R21EB028001.

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

## Appendix A. Detailed data information

Table 2 shows the data split of the four prostate MRI segmentation datasets used in this study. Table 3 shows detailed information regarding the dataset properties. The segmentation task in this study is to segment the whole prostate gland (including both the

prostate peripheral zone and the transition zone) from MRI images. The involved MR sequence type includes T2-weighted (T2W) series, apparent diffusion coefficient (ADC) series, proton density-weighted (PD-W) series, dynamic contrast-enhanced (DCE) series, and diffusion-weighted (DW) series. For the MSD dataset and NCI-ISBI dataset, the images are originally labeled with the prostate peripheral zone (PZ) and the transition zone (TZ). We combined these two regions to generate the segmentation mask of the whole prostate gland.

Table 2: Data split of the four prostate MRI segmentation datasets used in this study.

| Datasets | MSD[1] | NCI-ISBI[2] | PROMISE12[3] | PROSTATEx[4] | Total |
|---|---|---|---|---|---|
| Training (60%) | 19 | 48 | 30 | 122 | 219 |
| Validation (10%) | 3 | 8 | 5 | 20 | 36 |
| Testing (30%) | 10 | 24 | 15 | 62 | 111 |
| Total | 32 | 80 | 50 | 204 | 366 |

Table 3: Detailed data information of four client datasets used in this study.

| Datasets | Series type | Image size | | Spacing [mm] | | Manufacturers |
|---|---|---|---|---|---|---|
| | | X,Y | Z | X,Y | Z | |
| MSD | T2W/ADC | 256-384 | 11-24 | 0.60-0.75 | 3.00-4.00 | Not available |
| NCI-ISBI | T2W | 256-512 | 15-40 | 0.31-0.75 | 3.00-4.00 | Philips 1.5T + Siemens 3T |
| PROMISE12 | T2W | 256-512 | 15-54 | 0.27-0.75 | 2.20-4.00 | Not available |
| PROSTATEx | T2W/PD-W/DCE/DW | 320-640 | 18-27 | 0.30-0.60 | 3.00-4.50 | Siemens 3T |

---

1. http://medicaldecathlon.com

2. http://dx.doi.org/10.7937/K9/TCIA.2015.zF0vlOPv

3. http://promise12.grand-challenge.org

4. http://prostatex.grand-challenge.org

