# OpenReview forum: "Federated Cross Learning for Medical Image Segmentation"
_MIDL.io/2023/Conference — MIDL 2023 Poster_

### Official Review · Reviewer_ogX9 · 2023-02-01

**Confidence:** 5
**Preliminary Rating:** 3

**Summary:**

The author conduct a theoretical analysis on the FL algorithm to reveal the problem of model aggregation during training on non-iid
data. With the insights gained through the analysis, the author propose a simple yet effective method, federated cross learning (FedCross), to tackle this challenging problem. The experimental results also show that the proposed FedCross training strategy outperforms the mainstream FL methods on non-iid data.

**Strengths:**

The proposed method is different from the conventional FL methods that combine multiple individually trained local models on
a server node, FedCross sequentially trains the global model across different clients in a round-robin manner, and thus the entire training procedure does not involve any model aggregation steps.

**Weaknesses:**

My most critical concern is regarding the motivation. Because the non-iid problem is the basic problem in FL, which is well relieved by many federaed learning algorithms.  I totally can not find the novelty in this work, as the author just use some basic techiniques to deal with it. I think the author should clarify the motivation and novelty of this work.
My second concern is regarding the baselines. As there are many federated learning method on the medical image analysis. For example, Specificity-Preserving Federated Learning for MR Image Reconstruction CM Feng, Y Yan, W Shanshan, Y Xu, L Shao, H Fu IEEE Transactions on Medical Imaging. Li W, Milletarì F, Xu D, et al. Privacy-preserving federated brain tumour segmentation[C]//Machine Learning in Medical Imaging: 10th International Workshop, MLMI 2019, Held in Conjunction with MICCAI 2019,

**Deanonymize Review:**

yes

**Paper Type:**

validation/application paper

**Questions To Address In The Rebuttal:**

My most critical concern is regarding the motivation. Because the non-iid problem is the basic problem in FL, which is well relieved by many federaed learning algorithms.  I totally can not find the novelty in this work, as the author just use some basic techiniques to deal with it. I think the author should clarify the motivation and novelty of this work.
My second concern is regarding the baselines. As there are many federated learning method on the medical image analysis. For example, Specificity-Preserving Federated Learning for MR Image Reconstruction CM Feng, Y Yan, W Shanshan, Y Xu, L Shao, H Fu IEEE Transactions on Medical Imaging. Li W, Milletarì F, Xu D, et al. Privacy-preserving federated brain tumour segmentation[C]//Machine Learning in Medical Imaging: 10th International Workshop, MLMI 2019, Held in Conjunction with MICCAI 2019,

---

### Official Review · Reviewer_HWBA · 2023-02-04

**Confidence:** 3
**Preliminary Rating:** 4
**Recommendation:** Oral, Poster

**Summary:**

The paper proposes the federated learning method to train a deep learning model using isolated datasets from different centers. The method tackles the problem of non-iid data and features a round-robin training strategy and doesn't involve aggregation steps. In addition, the authors propose to train multiple models thus forming an ensemble. The method was tested using 4 public datasets and demonstrated excellent performance.

**Strengths:**

- The method is novel and innovative, and in combination with ensembling the RR strategy has advantages
- The paper is well-organized and easy to read. The method's design is well-motivated and supported by the analysis
- Solid design of the experiments, good results

**Weaknesses:**

I cannot pinpoint any major weakness, the method definitely has merit.
In the extended version, it would be interesting to see how well the performance scales with the number of centers, especially in the case when each center has little data. I expect the FedAvg and other methods to perform better in such cases, as the proposed method may face the problems of forgetting addressed in [1]

Also, it would interesting to see experiments on larger datasets and different tasks.

[1] Li, Zhizhong and Derek Hoiem. “Learning without Forgetting.” IEEE Transactions on Pattern Analysis and Machine Intelligence 40 (2016): 2935-2947.

**Deanonymize Review:**

no

**Paper Type:**

methodological development

**Questions To Address In The Rebuttal:**

In the rebuttal, it would be interesting to see how you expect the performance of the model to scale with the dataset size and with the growing number of participating centers. Would the proposed method work if each center has an incomplete or different set of annotations (labels)?

---

### Official Review · Reviewer_6H42 · 2023-02-04

**Confidence:** 5
**Preliminary Rating:** 4
**Recommendation:** Poster

**Summary:**

This paper presents a federated segmentation architecture (FedCross) where the global model is sequentially trained across different clients in a round-robin manner. This is proposed as an alternative to the standard federated averaging (FedAvg) model that combines the multiple locally trained models on the node server, and is shown to perform poorly when local data are not IID. This model is evaluated, based on the UNET backbone architecture on four public prostate MRI datasets. The FedCross model is shown to outperform local models trained on each dataset separately and to be in par with performance achieved with state of the art personalized FL models.

**Strengths:**

-This paper addresses the challenging federated segmentation task in non IID setting ie with local datasets that do not belong to the same distribution. Heterogeneity in the data may indeed impair performance of the standard federated averaging model.

-Performance of the proposed architecture is compared to those of some other state of the art FL methods

-The paper is well formatted, written and illustrated


**Weaknesses:**

- The proposed architecture is very similar to that proposed in https://www.nature.com/articles/s41598-020-69250-1, or in https://www.ncbi.nlm.nih.gov/pmc/articles/PMC8442829/#R25. The main difference is that the authors perform round robin selection instead of actual cyclic participation.

- The main limitation reported in cyclic incremental learning is that of catastrophic forgetting. I guess this should also happen with the proposed architecture.

-The authors do not provide any description, of the local datasets, which impairs estimating data heterogeneity. Since the proposed architecture specifically targets decentralized learning with non-IID data, such a description is needed as well as a fine analysis on how the observed performance variability reported in Table 1 relates to data heterogeneity. This also means, may be, to disentangle the impact of other factors, such as the unbalanced size of the local datasets.


**Deanonymize Review:**

no

**Detailed Comments:**

-The proposed architecture is not parallelizable, so that computation time is not comparable to standard FL. Please comment.

-The authors should provide details on the experimental analysis: first, the segmentation task should be specified, as well as the MR sequence type. I guess the task is that of segmenting the whole prostate gland (including both the peripheral and transition zone) based on T1 MRI. Please also append details on the databases, e.g in Table 2 of the Appendix; manufacturers, image size and resolution etc….Indeed, Difference in performance achieved on the different datasets may result from the dimension of the database, but also from different image patterns and texture achieved on scanners of different vendors or with different sequence parameters.

- Following my previous comment, Figure 3 outlines that the z-score normalisation does not succeed in erasing data shift for some local nodes. This might indicate that these different local datasets may differ by second order textural patterns.

-The authors should specify how the standard deviations in Table 1 were computed. Does it correspond to the standard deviation with regards to the DICE values computed on all test samples? The proposed ensemble model FedCrossEns, also enables deriving a standard deviation (uncertainty) from the mean value computed over the 4 models outputs, for each test sample? Please clarify how this standard deviation was accounted for and/or where it is reported in the paper.


**Paper Type:**

both

**Questions To Address In The Rebuttal:**

Please discuss the points i listed as 'weak' , regarding comparison with cyclic incremental learning and the issue of catastrophic forgetting. Also please address the questions of the 'comment' section.

---

### Meta-Review · Area_Chair_2Kny · 2023-02-25

**Recommendation:** Accept (Poster)
**Confidence:** 4

**Metareview:**

The authors propose an approach for federated learning (FL) to handle non-iid data for medical image segmentation. They provide a theoretical analysis to reveal a problem with training with model aggregation for non-iid data to justify the proposed approach, in which the global model is sequentially trained across different clients in a round-robin manner. The proposed approach is further combined with an ensemble learning mechanism. The methods are compared against common FL baselines on a prostate segmentation task.

Reviewers are in general agreement on the novelty and merit of the proposed approach, the strength of the experiments, and the nice and clear presentation of the paper. Concerns were brought up by multiple reviews regarding the problem of catastrophic forgetting and potential scalability of the approach to more clients, which was acknowledged by the authors as a potential limitation but is often shared by other FL approaches. Still given the strengths of the methods and experiments, I recommend acceptance of this work.